# Self-Reported Medication Use among Pregnant and Postpartum Women during the Third Wave of the COVID-19 Pandemic: A European Multinational Cross-Sectional Study

**DOI:** 10.3390/ijerph19095335

**Published:** 2022-04-27

**Authors:** Eva Gerbier, Guillaume Favre, Fatima Tauqeer, Ursula Winterfeld, Milos Stojanov, Alison Oliver, Anneke Passier, Hedvig Nordeng, Léo Pomar, David Baud, Alice Panchaud, Carla Meyer-Massetti, Michael Ceulemans

**Affiliations:** 1Materno-Fetal and Obstetrics Research Unit, Department “Woman-Mother-Child”, Lausanne University Hospital, 1011 Lausanne, Switzerland; guillaume.favre@chuv.ch (G.F.); milos.stojanov@chuv.ch (M.S.); leo.pomar@chuv.ch (L.P.); david.baud@chuv.ch (D.B.); alice.panchaud@chuv.ch (A.P.); 2Service of Pharmacy, Lausanne University Hospital and University of Lausanne, 1011 Lausanne, Switzerland; 3Pharmacoepidemiology and Drug Safety Research Group, Department of Pharmacy, PharmaTox Strategic Initiative, Faculty of Mathematics and Natural Sciences, University of Oslo, N-0315 Oslo, Norway; fatima.tauqeer@farmasi.uio.no (F.T.); h.m.e.nordeng@farmasi.uio.no (H.N.); 4Swiss Teratogen Information Service and Clinical Pharmacology Service, CHUV, 1066 Lausanne, Switzerland; ursula.winterfeld@chuv.ch; 5UK Teratology Information Service, Newcastle upon Tyne Hospitals NHS Foundation Trust and the UK Health Security Agency, Newcastle upon Tyne NE3 3HD, UK; alison.oliver12@nhs.net; 6Teratology Information Service, Pharmacovigilance Centre Lareb, ‘s Hertogenbosch, 5237 MH Hertogenbosch, The Netherlands; a.passier@lareb.nl (A.P.); michael.ceulemans@kuleuven.be (M.C.); 7Department of Child Health and Development, Norwegian Institute of Public Health, N-0213 Oslo, Norway; 8School of Health Sciences (HESAV), University of Applied Sciences and Arts Western Switzerland, 1011 Lausanne, Switzerland; 9Institute of Primary Health Care (BIHAM), University of Bern, CH-3012 Bern, Switzerland; carla.meyer-massetti@extern.insel.ch; 10Pharmacology & Toxicology, Department of General Internal Medicine, University Hospital of Bern, CH-3010 Bern, Switzerland; 11Department of Pharmaceutical and Pharmacological Sciences, KU Leuven, 3000 Leuven, Belgium; 12L-C&Y—KU Leuven Child & Youth Institute, 3000 Leuven, Belgium

**Keywords:** pregnancy, postpartum, pharmacoepidemiology, drug utilization, medication use, self-medication, chronic disease, COVID-19, pandemic, Europe

## Abstract

Information on medication utilization among pregnant and postpartum women during the pandemic is lacking. We described the prevalence and patterns of self-reported medication use among pregnant and postpartum women during the third wave of the pandemic (June–August 2021). An online questionnaire was distributed in five European countries between June–August 2021. Pregnant women or women who had delivered in the three preceding months, and ≥18 years old, could participate. The prevalence of overall medication use, self-medication, and changes in chronic medication use were determined. A total of 2158 women out of 5210 participants (41.4%) used at least one medication. Analgesics (paracetamol), systemic antihistamines (cetirizine), and drugs for gastric disorders (omeprazole) were the three most used classes. Anti-infectives were less prevalent than during pre-pandemic times. Antidepressants and anxiety related medication use remained similar, despite a higher prevalence of these symptoms. Self-medication was reported in 19.4% of women, and 4.1% of chronic medication users reported that they changed a chronic medication on personal initiative due to the pandemic. In conclusion, medication use patterns in our cohort were mostly similar to those of the first COVID-19 wave and the pre-pandemic period. More studies are needed to explore factors associated with self-medication and changes in chronic medication use due to the pandemic in this perinatal population.

## 1. Introduction

Since December 2019, the world has been struggling with the new severe acute respiratory syndrome coronavirus 2 (SARS-CoV-2), responsible for the COVID-19 pandemic and significant consequences on people’s daily lives and health [1,2,3,4]. Several vaccines are now available worldwide to prevent COVID-19 infection and progression to severe forms. Among the many medications that have been repurposed, a few have proved effective in treating the disease [5,6,7] while others have produced little to no benefits and were sometimes even harmful [8,9]. Hydroxychloroquine, in combination with azithromycin, was one striking example of this, showing no beneficial effects in several randomized clinical trials [10,11], but increased the likelihood of serious side effects [12]. In parallel, concerns related to the intake of certain commonly used medications, including non-steroidal anti-inflammatory drugs (NSAIDs) [13] and renin-angiotensin system medications [14], and a subsequent increased risk of severe COVID-19 have also been raised but have not been confirmed. This plethora of new and constantly changing information during the pandemic may have affected the behaviour of prescribers and patients regarding medication use.

In addition, the indirect effects of the pandemic may also have impacted medication practices, especially among vulnerable populations suffering from chronic diseases. For instance, studies have shown better adherence to asthma/COPD therapy during the pandemic, potentially due to patients’ fear of being more vulnerable to severe COVID-19 and to the strengthening of international society recommendations [15,16]. It has also been reported that some minorities experienced difficulties in accessing their treatment, in part due to loss of employment and living in communities with a high number of COVID-19 cases [17]. In addition, restricted or delayed access to health care is a factor that has impacted the management of chronic diseases. For example, an increased incidence of diabetic ketoacidosis cases between March 2020 and February 2021 have been reported in England among both type 1 and type 2 diabetes [18], suggesting poor glycemic control and delayed diagnosis. With regard to rheumatologic and autoimmune diseases, a switch from intravenous biologic drugs to subcutaneous treatments was observed as a consequence of the physicians’ preference to spare their patients from attending hospitals and thus lower their risk of infection [19]. Finally, the pandemic has taken a heavy toll on people’s mental health, and studies have shown different changes in the prescriptions of psychotropic medications (e.g., antipsychotics, antidepressants, benzodiazepines) during this period, with variations between medication classes, populations, and period of the pandemic. For instance, an increase in antidepressant prescriptions among all UK residents was observed between January and August 2020 compared to the years 2016–2019 [20], while a decrease was observed among male patients in Portugal between March 2020 and March 2021 [21]. In the latter study, anxiolytics, hypnotics, and sedative prescriptions decreased among elderly females over the same period, while prescriptions of antipsychotics, benzodiazepines, and antidepressants increased among residents of nursing homes in Canada after the onset of the pandemic [22].

While evidence on medication practices and utilization patterns in the general population during the pandemic is continuously growing, it remains scarce among pregnant and postpartum women. A multinational study performed across five European countries at the end of the first wave of the pandemic (i.e., June–July 2020) reported on the use of medication in this specific population [23]. Results showed a high prevalence of self-reported medication use (60%), with similar medication classes being frequently used during the pandemic as before, except for the level of antibiotic use. Moreover, 20% of women declared that they were more likely to take medication because of the pandemic. However, the actual prevalence of self-medication during pregnancy in relation to the pandemic and any changes in chronic medication use related to the COVID-19 pandemic were not studied. Meanwhile, the pandemic has continued its course, and several new waves have passed, possibly impacting medication utilization patterns in this population. Thus, the aim of this study was two-fold: first, we aimed to describe the prevalence and patterns of self-reported medication use among pregnant and postpartum women during the third wave of the pandemic (June–August 2021); second, we aimed to assess changes in chronic medication use due to COVID-19 and associated risk factors.

## 2. Materials and Methods

### 2.1. Design and Study Population

This study is a cross-sectional drug utilization study. It is part of a multinational, European, COVID-19 project aimed at determining the impact of the third wave of the pandemic on pregnant and postpartum women’s medication use, vaccine acceptance, mental health, and perinatal/birth experiences. An online, anonymous questionnaire was distributed in Norway, Belgium, Switzerland, the Netherlands, and the United Kingdom between 10 June and 22 August 2021 (see Appendix A for the English version). The questionnaire was adapted from the first round of the COVID-19 project in 2020 [2,23,24]. Pregnant women and women who had given birth in the three months preceding the survey and who were at least 18 years old, were eligible to participate. Given the anonymous nature of the data collected, ethics and privacy approval were not required in the participating countries; in Belgium, approval was obtained from EC Research UZ/KU Leuven (S63966, 27 May 2021). Participants provided online informed consent before accessing the questionnaire.

For the first aim, we included the entire study population. For the second aim, we restricted the population to women who reported a change or no change in the use of a chronic medication during the three preceding months, excluding those who reported not having used any chronic medication (see flowchart in Figure 1).

### 2.2. Data Collection

The questionnaire was accessible online and promoted through commonly used pregnancy/maternal websites, forums, and social media. Details on the recruitment tools and availability of the questionnaire in each country are provided in Appendix A, as well an overview of the infection rates and imposed regulations in each country during the study period in Appendix A. The national study coordinators translated the English version of the modified survey into the respective national language(s) of the participating countries (i.e., Norwegian, Dutch, French, German, and Italian).

### 2.3. Use of Medication

All pregnant and postpartum participants were asked whether they had used any medicinal product(s) in the three months preceding the survey including both “prescription and non-prescription medicines, folic acid, and herbal medicines”. Information on the name of the product(s) used as well as the disease(s) and symptom(s) for which the product(s) was/were used were retrieved. Women were also asked about the occurrence and type of ‘self-medication’ during pregnancy or since delivery, defined as “starting a medication on personal initiative without advice from a health care professional (HCP)”. Potential reasons were explored by using multiple-choice questions. Third, any changes in the use of chronic medications on personal initiative due to the COVID-19 pandemic were investigated through yes or no and open-ended questions, exploring the names of the changed medications and the reasons for changing.

National study coordinators classified each medicinal product into six different groups including “medications”, “iron”, “folic acid”, “multivitamins”, “omega-3 fatty acids”, and “other products”. The latter included probiotics, herbal products, vitamin C, calcium, and all other health products that did not belong to any other group. Iron, folic acid, multivitamins, omega-3 fatty acids, and other products are referred to as “supplements/vitamins” in the text. Medications were coded according to the Anatomical Therapeutic Chemical (ATC) classification system [25] into the 5th ATC level whenever possible, otherwise into the most precise corresponding ATC level (2nd–4th). In cases where the product contained more than one active substance that did not correspond to a specific ATC code, each substance was coded separately. When the route of administration was not provided, it was coded as the one most commonly used in the context of pregnancy.

### 2.4. Covariates

In addition to medication use, the survey collected information on the participants’ socio-demographic characteristics (i.e., country, age, relationship status, professional status, working in health care or not, highest level of education), their current pregnancy (i.e., gestational age, planned pregnancy or not) or postpartum status (i.e., infant’s age, currently breastfeeding or not, any breastfeeding experience before the pandemic), their general health (i.e., presence of a chronic disease: asthma, allergy, hypothyroidism, cardiovascular diseases, diabetes, depression, anxiety, epilepsy, rheumatoid arthritis, inflammatory bowel disease, any other disease), the presence of depressive and anxiety symptoms (based on the Edinburgh Depression Scale (EDS) [26], and the Generalized Anxiety Disorder Assessment (GAD-7) [27]), and substance use (i.e., smoking, alcohol consumption). Information specifically related to the pandemic was also collected such as a potential or confirmed COVID-19 infection, the severity of the infection, the infection of a close family member, their vaccine status (dichotomized as being vaccinated or willing to vs. unwilling to be vaccinated), the perception of anti-pandemic measures as an infringement to individual freedom (dichotomized as yes/no), and trust in various sources of information regarding the pandemic (either the government or health authorities, at least one type of HCP (i.e., general practitioner, pharmacist, midwife, or obstetrician), family or friends, the media or Internet).

### 2.5. Data Analyses

Aim 1: Descriptive analyses were used to calculate the prevalence of any medicinal product (including medications and supplements/vitamins) and any medication use (excluding supplements/vitamins) during pregnancy and postpartum as well as the mean number of medications used per woman. The prevalence of the ten most used medication classes (ATC level 2) was determined, along with the most commonly used individual medications (ATC level 5) within each class. The prevalence of self-medication per class, and potential reasons for self-initiating medication were described as absolute numbers and percentages.

Aim 2: The prevalence of changes in the use of a chronic medication due to COVID-19 was determined. Reasons for changes in chronic medication use were qualitatively assessed using an inductive, thematic approach, grouping the answers into thematic key concepts. An exploratory case-control analysis was further conducted to determine risk factors associated with changes in chronic medication due to COVID-19. Pregnant and postpartum women who answered positively to the following question: “Did you change the use of a chronic medicine on your own initiative due to the COVID-19 pandemic (e.g., stopped using it, used less or used more of the medicine)?” were defined as cases. Pregnant and postpartum women who answered negatively to the latter question were defined as controls. Women who did not answer the question or answered “I don’t use a chronic medicine” were excluded from the analysis. Risk factors identified in the general population for changes in chronic medication during the pandemic were tested in a univariable logistic regression such as (1) disruption in access to health care services [17,18,19], which corresponded to either having experienced a change in the schedule of persons providing care (midwife, gynecologist) or cancellation/reduction of an appointment (dichotomized as yes/no); (2) fear of contracting a severe form of COVID-19 (dichotomized as yes/no); and (3) perceiving COVID-19 as more severe or risky during pregnancy/breastfeeding compared to the general non-pregnant/non-breastfeeding population (categorized as more severe/risky, very risky, extremely risky, and not more severe/not very risky/not risky at all) [28,29]. Depressive and anxiety symptoms, both commonly associated with poor medication adherence [30,31] and likely worsened during the pandemic, were also tested. Finally, potential risk factors specific to the pandemic situation were analyzed. Covariates with a *p* ≤ 0.25 were included in the multivariable logistic regression. Standardized difference was used to assess imbalances in baseline characteristics between women who changed a chronic medication due to COVID-19 and women who did not. Covariates with a standardized difference equal to or greater than 0.20 were also included in the final adjusted model. Multicollinearity was checked using the variance inflation factor (VIF), considering 10 as a threshold for multicollinearity. Results are presented as crude odds ratio and adjusted odds ratio with 95% confidence intervals. Statistical analyses were performed with STATA SE version 17 (StataCorp, College Station, TX, USA).

## 3. Results

### 3.1. Characteristics of the Population

Overall, 5210 women participated in the survey, encompassing 3411 pregnant women and 1799 women who had delivered in the three months preceding the survey. Most women were recruited in Norway (67.0%, 3489/5210), followed by Belgium (11.4%, 595/5210), the United Kingdom (7.9%, 410/5210), Switzerland (7.4%, 386/5210), and the Netherlands (6.3%, 330/5210). Most participants were in a relationship (88.5%, 4611/5210) and were professionally active (80.0%, 4167/5210), with approximately one out of four women (26.2%, 1366/5210) working in health care. A chronic disease was reported by 34.9% (1190/3411) and 26.5% (476/1799) of pregnant and post-partum women, mainly allergies (18.6%, 634/3411 and 12.8%, 230/1799 of pregnant and post-partum women, respectively) and asthma (5.7%, 195/3411 and 4.1%, 74/1799). Hypothyroidism was the third most common chronic disease during pregnancy (4.6%, 157/3411), while cardiovascular diseases were the third most common during postpartum (2.6% 46/1799). Pregnant women were nulliparous in 53.7% (1831/3411) and in their third trimester in 52.8% of the cases (1802/3411). The pregnancy was planned in 78.4% of cases. Most postpartum women were breastfeeding when filling out the survey (89.2%, 1604/1799), and 61.3% (1102/1799) of their infants were between six and 12 weeks old. Of all participants, 5.1% (263/5210) had tested positive for a SARS-CoV-2 infection at any time since the beginning of the pandemic, and 7.4% (383/5210) were symptomatic but not confirmed with a test including 1.7% (88/5210) who had either presented long-term symptoms or had been hospitalized.

Overall, 93.1% (4851/5210) of participants including 93.6% (3194/3411) of pregnant and 92.1% (1657/1799) of postpartum women reported having used a “medicinal product” (including medications and supplements/vitamins) in the last three months. This corresponded to 32.5% (1134/3489) of Norwegian women, 65.5% (390/595) of Belgian women, 57.6% (236/410) of British women, 61.4% (237/386) of Swiss women, and 48.8% (161/330) of Dutch women. Participant characteristics according to medication use (excluding iron, folic acid, multivitamins, omega-3 fatty acids, and other products) in the last three months are presented in Table 1. The use of supplements/vitamins (i.e., iron, folic acid, multivitamins, omega-3 fatty acids, and other products) by country is described in Appendix A.

### 3.2. Overall Medication Use

Overall, 41.4% (2158/5210) of women reported having used at least one medication (excluding supplements/vitamins) in the three preceding months (i.e., 29.2% (997/3411) of pregnant and 64.5%, 1161/1799 of postpartum women). The average number of medications reported was 1.9 (min–max: 1–8) among pregnant women and 2.2 (min–max: 1–9) among postpartum women. Overall, analgesics were the most reported medication class among both pregnant (13.8%, 470/3411) and postpartum women (44.0%, 791/1799), mainly represented by paracetamol in both groups (10.4% and 35.0%, respectively). Pregnant women further reported the frequent use of systemic antihistamines (7.3% including cetirizine 1.8%) and drugs for gastric related disorders (7.1% including omeprazole 1.5%). Postpartum women mainly reported NSAIDs (16.7% including ibuprofen 14.3%) and systemic antihistamines (12.1% including cetirizine 4.1%). Table 2 provides an overview of the ten most reported ATC classes and individual medications within each class among pregnant and postpartum women, in order of descending prevalence for pregnant women (see Appendix A for an overview by country). Thus, the medication classes and individual medications among post-partum women do not necessarily correspond to the ten most frequent ones within this group. For instance, NSAIDs do not appear in this table despite being the second most used class among postpartum women (16.7%).

### 3.3. Self-Medication

In total, 18.0% (478/2662) of pregnant women and 22.5% (291/1292) of postpartum women reported having started a medication on their own initiative during pregnancy or since delivery without the advice of a HCP. The most frequently reported medications by pregnant women as self-medication were paracetamol (32.8%, 157/478), alginic acid (5.2%, 25/478), and ordinary salt combinations as antacids (combinations of calcium, aluminum, and magnesium) (4.6%, 22/478). Those reported by postpartum women were paracetamol (76.6%, 223/291), ibuprofen (29.2%, 85/291), and cetirizine (4.1%, 12/291). With respect to potential reasons for self-medication, pregnant women and postpartum women self-medicated mostly because: “a HCP previously told me that the medication was safe during pregnancy/breastfeeding” (289/478, 60.5% and 205/291, 70.4%); “I read on the Internet that the medication is safe during pregnancy/breastfeeding” (232/478, 48.5% and 135/291, 46.4%) and “A friend/family member previously told me that the medication is safe during pregnancy/breastfeeding” (87/478, 18.2% and 21/291, 7.2%). Reasons related to the COVID-19 pandemic were mentioned by 4.6% (22/478) of pregnant and 8.2% of postpartum women (24/291) including difficulties to access a HCP, and unwillingness to attend a health care facility/use time from a HCP.

### 3.4. Changes in Chronic Medication Use Related to COVID-19

Figure 1 shows the number of pregnant and post-partum women who changed the use of a chronic medication among chronic medication users.

**Figure 1 ijerph-19-05335-f001:**
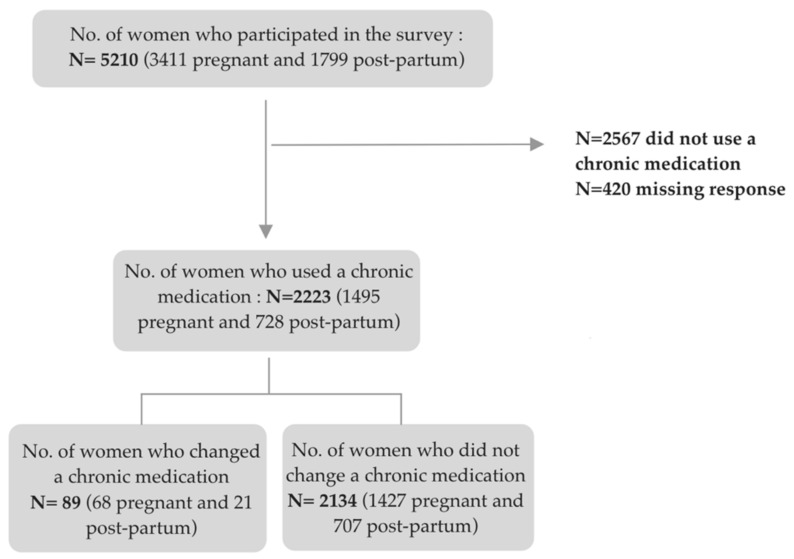
Flowchart of the study participants.

Among chronic medication users (*n* = 2223), 4.1% (*n* = 89) (i.e., 4.6% and 2.9% of pregnant and postpartum women, respectively) indicated having changed a chronic medication on their own initiative due to the pandemic. Table 3 presents the characteristics of chronic medication users according to any change in medication use on their own initiative due to the COVID-19 pandemic.

An overview of the ten most reported ATC classes of chronic medications changed on the women’s own initiative due to the pandemic is shown in Appendix A. Analgesics (1.2%, 26/2223), systemic antihistamines (0.9%, 19/2223), and anti-inflammatory drugs (0.4%, 9/2223) were the three medication classes women mostly changed on their own initiative due to the pandemic. The most changed medications among pregnant women were paracetamol (0.7%, 10/1495), cetirizine (0.5%, 8/1495), ibuprofen (0.4%, 6/1495), and sumatriptan (0.4%, 6/1495). Among post-partum women, these were budesonide (0.3%, 2/728) and desloratadine (0.3%, 2/728). Reasons for changing chronic medications can be summarized into five concepts/groups: (1) “Limited access to a HCP/health care service”; (2) “Fear of contracting the disease/preventive measure”; (3) “Stress and uncertainty”; (4) “Fear of public stigmatization”; and (5) “Vaccine-related side-effects” (see Appendix A).

### 3.5. Factors Associated with Changes of Chronic Medication Due to COVID-19

Table 4 illustrates the associations of chronic medication changes due to COVID-19 among chronic medication users. In the multivariable analysis, depressive symptoms (EDS ≥ 13) were negatively associated with the change in a chronic medication due to COVID-19 (aOR 0.5 [0.3–0.9]) and mild symptoms of anxiety (aOR 2.3 [1.4–3.8]) were positively associated. A positive trend could be observed among women with moderate (aOR 2.0 [1.0–4.2]) and severe anxiety (aOR 2.0 [0.8–5.1]) as well as among women who trusted their family or friends (aOR 1.6 [1.0–2.8]) as a source of information for COVID-19.

## 4. Discussion

### 4.1. Main Findings

This multinational cross-sectional study investigated the prevalence and patterns of self-reported medication use during the third wave of the COVID-19 pandemic among pregnant and postpartum women. Along with the findings from a previous study conducted at the end of the first wave [23], this study provides insight into the utilization of medication in this population during the pandemic. Moreover, the current study provides unique information on self-medication during pregnancy and changes in the use of chronic medications due to the pandemic, which has not been previously reported.

Overall, the prevalence of self-reported medication use in this study compared to the first wave was lower among pregnant women (31.2% vs. 59%) but higher among postpartum women (70.1% vs. 56%). Among pregnant women, this may be explained by the higher proportion of Norwegian respondents in this survey (67%) compared to the first one (34%), who tend to use less medication than pregnant women living in other countries [23]. However, in both surveys, the observed prevalence rates of medication use in pregnancy are likely to be an underestimation as women were asked to report on medication use during the last three months only and not during their entire pregnancy. In comparison, Lupatelli et al. reported that 81.2% of pregnant women and new mothers declared having used a medication at some point during their entire pregnancy through an online questionnaire distributed in 18 countries [32]. Regarding postpartum women, the higher prevalence of medication use observed in this study could be explained by the inclusion of all postpartum women, regardless of whether they were breastfeeding or not, who had given birth in the previous three months, compared to the previous study, in which only women who had breastfed their child in the three months preceding the survey were included.

Analgesics, systemic antihistamines, and drugs for gastric related disorders were the three most used medication classes among pregnant women both during the first (28%, 15%, 11%, respectively) and third wave of the pandemic (15%, 8%, 8%, respectively), in line with pre-pandemic observations [32]. However, similarly to the first wave (3%), the self-reported use of systemic antibiotics was less prevalent among pregnant women (2%) than in pre-pandemic reports. Indeed, 14% of Lupatelli et al.’s pregnant participants reported having used a systemic antibiotic [32], and at least 26% of women were prescribed an antibiotic at some point in pregnancy, according to a recently published Swiss claims database cohort [33]. Although our prevalence of systemic antibiotic use may be underestimated due to questioning only the three preceding months, the use/dispensing of systemic antibiotics was equally frequent in each pregnancy trimester in both studies [32,33]. Thus, a three-month study period should provide a good reflection of antibiotic use during pregnancy [32,33]. Similar decreases in antibiotics use have also been observed in the general population during the pandemic [34], suggesting a reduced incidence of other infectious diseases due to preventive sanitary measures against COVID-19.

The use of antidepressants was reported by 1.9% of all women in our study, which is similar to estimates calculated during the first wave (2%) and the pre-pandemic era (2%) [23,32,35]. In parallel, 18.3% of all participants experienced potential depressive symptoms (EDS ≥13), which was higher than pre-pandemic estimates [36,37,38], with certain exceptions [39]. An even higher prevalence of depressive symptoms during the pandemic was observed among pregnant women in two survey-based studies, the first one including 1987 participants in Canada (EDS ≥ 13: 37%) [40], and the second, 260 participants in Turkey (EDS >13: 35%) [41]. These studies were both conducted at the beginning of the pandemic when uncertainty regarding the severity of the virus was at its highest and strict measures including social isolation and lockdown were in place, partly explaining this very high prevalence. Nonetheless, it appears that the pandemic may have heightened depressive symptoms in this vulnerable population, which stresses the importance of maintaining screening and counselling of the mental health of pregnant women, even more so perhaps during pandemic times. Similarly, the use of anxiety-related medications (1.5%) has remained close to pre-pandemic numbers (varying between 1.5% and 1.8% across pregnancy trimesters) [32], while more than half (52.4%) of participants presented symptoms of anxiety on the GAD-7 scale compared to 22.9% and 17.8% of pregnant and postpartum women, respectively, before the pandemic [42]. This could potentially reflect the underdiagnosis and undertreatment of these symptoms in the perinatal population during the pandemic.

Self-medication in our study population (19.4%) was low compared to pre-pandemic observations (66.9%) [32]. Once again, our prevalence do not reflect the total medication use throughout pregnancies, given the time restriction applied to medication reporting in our survey. Although most pregnant and postpartum women self-medicated because they were previously told that the medication was safe by a HCP, a large proportion declared doing so after having read information on the Internet or having received this information from a friend/family member, stressing the importance of providing appropriate, reliable sources of information to this population. This is perhaps even more important during pandemic times since difficulties or unwillingness to access HCPs have been shown to impact the physicians’ prescriptions [19], or patients’ medication adherence to chronic treatment [15,16]. In our study, difficulties or unwillingness to access HCPs were mentioned by almost 6% of women who self-medicated.

The proportion of women having changed the use of a chronic medication based on personal initiative due to COVID-19 was relatively low (4.1%). In fact, this proportion is likely an overestimation of the actual situation as some women reported “non-chronic” medications such as paracetamol. The most changed “chronic” medications reported by women were analgesics and antihistamines. The identification of these classes is, however, not surprising given that both classes are typically the most used ones during pregnancy and postpartum [32,33], which was also the case in our study population. Although asthma medications were not among the ten most used classes in our study, they were the fourth most changed class due to COVID-19. Increased use of asthma medications, along with nasal preparations and anti-allergic medications, to either prevent infection or for fear of public stigmatization, were cited among reasons for changes in chronic medications in our study. This finding is consistent with other literature reports stating an increase in asthma medication prescriptions [43] and medication adherence [15] during the pandemic. In parallel, a peak on asthma Internet searches was observed at the beginning of the pandemic [44], possibly mirroring asthma patients’ heightened worry regarding their vulnerability both towards contracting the disease and developing severe COVID-19. The impact of online information on asthma patients’ treatment during the pandemic is unknown, but it is likely that patients followed official guideline recommendations to be supplied with enough medication and to take the medication as prescribed [45]. In our study, we did not find an association between trust in the Internet/the media or any other source of information for COVID-19 and self-reported changes in chronic medication use on personal initiative, possibly due to the small sample size, although a positive trend between trust in family/friends and changes in chronic medication was observed.

Finally, a negative association between depressive symptoms and self-initiated changes in chronic medication use was observed. This finding is coherent with the stable prevalence of antidepressants across pre-pandemic times, the first wave and our study, despite a high prevalence of depressive symptoms. In contrast, women with mild symptoms of anxiety were more likely to personally change the use of a chronic medication due to COVID-19. A positive trend was also observed for women with moderate and severe anxiety symptoms. Although the direction of change was not specified in the survey question, some women likely intensified their treatment while others may have stopped or decreased it, potentially explaining the overall lack of change in anxiety-related medication use in our cohort compared to before the pandemic. Indeed, certain women who mainly managed to control their anxiety with psychological therapy and limited use of medications before the pandemic may have required an increase in anxiety-related medications due to restricted time with their therapist [46] or exacerbation of their symptoms [47,48]. On the other hand, other women suffering from anxiety may have found relief in the sanitary measures leading to less social contact, less transport, and working from home, and thus decreased or stopped their pharmacological treatment.

### 4.2. Strengths and Limitations

To our knowledge, this study provides a unique overview of medication use during the third COVID-19 pandemic wave among a large number of pregnant and postpartum women living across several European countries. Data collection and analysis were uniform across countries, allowing for comparison and pooling of the results. National coordinators’ knowledge of trade names and medication practices in each country also increased the validity of the results. The use of an anonymous questionnaire may have encouraged women to disclose all the products used, resulting in (more) complete data collection.

Nevertheless, our results should be interpreted in light of certain limitations. First, we observed an over-representation of Norwegian women, who tend to use less medication [32] compared to other countries. Women in our study were also younger and had a higher educational level compared to national birthing populations (see Appendix A), almost one third was working in health care, more than half of the pregnant women were in their third trimester, and approximately 90% of postpartum women were breastfeeding. Older maternal age has been associated with higher medication use [49], and thus our prevalence rates of medication use may be an underestimation. However, this may be counteracted by the substantial proportion of women working in health care who tend to use more medication during pregnancy [32]. Second, women were asked to report only on the three preceding months, and therefore we did not capture a complete picture of medication use during the entire pregnancy. Third, the definition of “chronic” medication use would have required additional clarification, since some women included certain “non-chronic” medications in their answers. Fourth, the direction of change (i.e., whether women stopped, increased, or decreased the use of their medication when they reported a change) remains unknown and requires further investigation. Fifth, the regression analysis aimed at identifying factors associated with self-initiated changes in chronic medication use may have been underpowered given the small sample of women included in this analysis. Finally, the cross-sectional nature of this study precludes any causal attribution to the factors we assessed regarding changes in chronic medication due to the pandemic.

## 5. Conclusions

Medication use patterns in our perinatal cohort were similar to those of the first COVID-19 wave and the pre-pandemic period, except for the lower level of systemic antibiotic use, likely reflecting the impact of preventive sanitary measures on the occurrence of other infectious diseases. A stable prevalence of antidepressants and anxiety-related medications compared to the first wave and pre-pandemic was observed, despite a high(er) proportion of women reporting symptoms of depression and anxiety. Self-medication was reported by one out of five women in our study population. The main reason for self-medication was that women had previously received advice from a HCP, but a critical proportion either relied on the Internet or their friends or family for advice. Finally, only a few percent of women indicated having changed the use of a chronic medication on personal initiative due to the COVID-19 pandemic. Among those, a positive trend was observed between women who trusted their friends/family and self-initiated change in chronic medication use due to COVID-19. More studies are needed to explore perinatal medication use throughout different pandemic waves and in different settings as well as to provide insights into the risks and protective factors associated with self-medication and changes in chronic medication use due to the pandemic in this population.

## Figures and Tables

**Table 1 ijerph-19-05335-t001:** Characteristics of the participants according to any medication use in the last three months (*n* = 5210).

	All Pregnant Women (*n* = 3411)	Pregnant Women Who Used at Least One Medication during the Three Previous Months (*n* = 997)	All Post-Partum Women (*n* = 1799)	Postpartum Women Who Used at Least One Medication during the Three Previous Months (*n* = 1161)
	N (%)	N (%)	N (%)	N (%)
**Baseline characteristics**				
Country				
Norway	2376 (69.7)	408 (40.9)	1113 (61.9)	726 (62.5)
Belgium	360 (10.6)	229 (23.0)	235 (13.1)	161 (13.9)
United Kingdom	290 (8.5)	166 (16.7)	120 (6.7)	70 (6.0)
Switzerland	210 (6.2)	113 (11.3)	176 (9.8)	124 (10.7)
The Netherlands	175 (5.1)	81 (8.1)	155 (8.6)	80 (6.9)
Maternal age (years)				
18–30	1374 (40.3)	387 (38.8)	675 (37.5)	473 (40.7)
31–40	1707 (50.0)	544 (54.6)	897 (49.9)	631 (54.4)
>40	68 (2.0)	35 (3.5)	44 (2.4)	31 (2.7)
Relationship status				
Married/cohabiting/partner	3092 (90.1)	941 (94.4)	1595 (88.7)	1120 (96.5)
Single	57 (1.7)	25 (2.5)	21 (1.2)	15 (1.3)
Professional status *				
Professionally active	2799 (82.1)	869 (87.2)	1430 (79.5)	1005 (86.6)
Not professionally active	348 (10.2)	96 (9.6)	176 (9.8)	122 (10.5)
Working in health care				
Yes	906 (26.6)	327 (32.8)	486 (27.0)	348 (30.0)
No	1879 (55.1)	542 (54.4)	947 (52.6)	659 (56.8)
Level of education				
Low (primary school)	73 (2.1)	16 (1.6)	34 (1.9)	22 (1.9)
Medium (high school)	540 (15.8)	157 (15.7)	290 (16.1)	773 (66.6)
High (more than high school)	2516 (73.8)	781 (78.3)	1279 (71.1)	328 (28.3)
Smoked during pregnancy **				
Yes	43 (1.3)	24 (2.4)	19 (1.1)	13 (1.1)
No	3106 (91.1)	942 (94.5)	1597 (88.8)	1122 (96.6)
At least one chronic disease				
Yes	1190 (34.9)	467 (46.8)	476 (26.5)	456 (39.3)
No	2221 (65.1)	530 (53.2)	1323 (73.5)	705 (60.7)
**Reproductive health**				
Parity				
Nulliparous	87 (3.0)	528 (53.0)	N/A	N/A
Multiparous	1580 (46.3)	469 (47.0)	N/A	N/A
Planned pregnancy				
Yes	2673 (78.4)	831 (83.4)	N/A	N/A
No	208 (4.4)	57 (5.7)	N/A	N/A
No, but it was not unexpected	465 (10.4)	109 (10.9)	N/A	N/A
Gestational weeks (GW)				
First trimester (<14 GW)	393 (11.5)	121 (12.1)	N/A	N/A
Second trimester (>14 and <28 GW)	1151 (33.7)	371 (37.2)	N/A	N/A
Third trimester (28 GW-end of pregnancy)	1802 (52.8)	505 (50.7)	N/A	N/A
Infant’s age				
≤6 weeks	N/A	N/A	674 (37.5)	447 (38.5)
6–12 weeks	N/A	N/A	1102 (61.3)	714 (61.5)
Currently breastfeeding				
Yes	N/A	N/A	1604 (89.2)	1050 (90.4)
No	N/A	N/A	172 (9.6)	111 (9.6)
Any breastfeeding experience before the pandemic				
Yes	N/A	N/A	660 (36.7)	437 (37.6)
No	N/A	N/A	942 (52.4)	613 (52.8)
**COVID-19 related factors**				
Coronavirus infection status ***				
Positive test	163 (4.8)	63 (6.3)	100 (5.6)	58 (5.0)
Symptomatic but not tested	224 (6.6)	85 (8.5)	159 (8.8)	102 (8.8)
Negative test or no symptoms	2958 (86.7)	849 (85.1)	1501 (83.4)	259 (22.3)
Infection severity				
No/mild symptoms	128 (3.8)	54 (5.4)	97 (5.4)	57 (4.9)
Moderate symptoms	185 (5.4)	68 (6.8)	72 (4.0)	45 (3.9)
Hospitalization/long term symptoms	70 (2.1)	26 (2.6)	18 (1.0)	11 (1.0)
Family member infection				
Yes	233 (6.8)	124 (12.4)	153 (8.5)	125 (10.8)
No	1262 (37.0)	488 (49.0)	575 (32.0)	487 (42.0)

Results are expressed as absolute numbers (%). N/A = question was not applicable. * At the time of survey completion or before maternity leave. ** Smoking during pregnancy for both pregnant and post-partum. *** Anytime during the pandemic. Numbers may not add up due to missing values; Missing values for pregnant women: maternal age, relationship status, *n* = 262 (7.7%), professional status, *n* = 264 (7.7%), working in health care, *n* = 626 (18.4%), level of education, *n* = 282 (8.3%), smoking during pregnancy, *n* = 262 (7.7%), parity, *n* = 1744 (51.2%), planned pregnancy, *n* = 65 (1.9%), gestational age, *n* = 65 (1.9%), coronavirus infection status, *n* = 66 (1.9%), infection severity, *n* < 5, family member infected with COVID-19, *n* = 71 (2.1%), Missing values for postpartum women: maternal age, relationship status, *n* = 183 (10.2%), professional status, *n* = 193 (10.7%), health care worker, *n* = 366 (20.3%), education level, *n* = 196 (11.0%), smoking in postpartum, *n* = 183 (10.2%), infant’s age, *n* = 23 (1.3%), currently breastfeeding, *n* = 23 (1.3%), previous breastfeeding experience, *n* = 197 (11.0%), coronavirus infection status, *n* = 39 (2.2%), infection severity, *n* = 72, (4%), family member infected with COVID-19, *n* = 41 (2.3%).

**Table 2 ijerph-19-05335-t002:** The ten most reported ATC classes and individual medications within each class among pregnant and postpartum women.

	Medication Class (ATC)	Pregnant Women (*n* = 3411)	Postpartum Women (*n* = 1799)	Medication Class (ATC)
		N (%)	N (%)	
1	Analgesics (N02)	470 (13.8)	791 (44.0)	Analgesics (N02)
	Paracetamol (N02BE01)	355 (10.4)	630 (35.0)	Paracetamol (N02BE01)
2	Antihistamines for systemic use (R06)	248 (7.3)	243 (12.0)	Antihistamines for systemic use (R06)
	Cetirizine (R06AE07)	62 (1.8)	73 (4.1)	Cetirizine (R06AE07)
3	Drugs for acid related disorder (A02)	243 (7.1)	214 (11.9)	Drugs for acid related disorder (A02)
	Omeprazole (A02BC01)	50 (1.5)	87 (4.8)	Not specified
4	Antithrombotic agents (B01)	116 (3.4)	64 (3.6)	Antithrombotic agents (B01)
	Acetylsalicylic acid (B01AC06)	75 (2.2)	18 (1.0)	Enoxaparin (B01AB05)
5	Medications for constipation (A06)	90 (2.6)	167 (9.3)	Medications for constipation (A06)
	Lactulose (A06AD11)	36 (1.1)	52 (2.9)	Lactulose (A06AD11)
6	Medications for functional gastrointestinal disorders (A03)	85 (2.5)	34 (1.9)	Medications for functional gastrointestinal disorders (A03)
	Metoclopramide (A03FA01)	25 (0.7)	19 (1.1)	Not specified
7	Medications for obstructive airway diseases (R03)	79 (2.3)	70 (3.9)	Medications for obstructive airway diseases (R03)
	Salbutamol (R03AC02)	24 (0.7)	18 (1.0)	Salbutamol (R03AC02)
8	Thyroid therapy (H03)	74 (2.2)	68 (3.8)	Thyroid therapy (H03)
	Levothyroxine sodium (H03AA01)	73 (2.1)	68 (3.8)	Levothyroxine sodium (H03AA01)
9	Antibacterials for systemic use (J01)	59 (1.7)	168 (9.3)	Antibacterials for systemic use (J01)
	Not specified	27 (0.8)	87 (4.8)	Not specified
10	Sex hormones and modulators of the genital system (G03)	55 (1.6)	32 (1.8)	Sex hormones and modulators of the genital system (G03)
	Desogestrel (G03AC09)	15 (0.)	16 (0.9)	Desogestrel (G03AC09)

Results are expressed as absolute numbers (%).

**Table 3 ijerph-19-05335-t003:** Characteristics of chronic medication users (*n* = 2223) according to any change in medication use on personal initiative due to the COVID-19 pandemic.

	Chronic Medication Users Who Changed a Chronic Medication (*n* = 89)	Chronic Medication Users Who Did Not Change a Chronic Medication (*n* = 2134)	Standardized Differences *
	N (%)	N (%)	
Country of response			**0.39**
Norway	67 (75.3)	1361 (63.8)	
Belgium	10 (11.2)	275 (12.9)	
United Kingdom	6 (6.7)	179 (8.4)	
Switzerland	<5	188 (8.8)	
Netherlands	5 (5.6)	131 (6.1)	
Maternal age			0.18
18–30	32 (36.0)	888 (41.6)	
31–40	50 (56.2)	1178 (55.2)	
>40	5 (5.6)	54 (2.5)	
Relationship status			0.16
Married/cohabiting/partner	3 (3.4)	40 (1.9)	
Single	84 (94.4)	2080 (97.5)	
Professional status			0.16
Professionally active	79 (88.8)	1841 (86.3)	
Not professionally active	8 (9.0)	273 (12.8)	
Working in health care			0.09
Yes	51 (57.3)	1210 (56.7)	
No	28 (31.5)	626 (29.3)	
Education level			0.11
Low (primary school)	3 (3.4)	44 (2.1)	
Medium (high school)	14 (15.7)	387 (18.1)	
High (more than high school)	70 (78.7)	1672 (78.4)	
At least one chronic disease			**0.23**
Yes	63 (70.8)	1277 (59.8)	
No	26 (29.2)	857 (40.2)	
Smoked during pregnancy or since delivery			0.18
Yes	3 (3.4)	32 (1.5)	
No	84 (94.4)	2088 (97.8)	

* A standardized difference equal or greater than 0.20 was considered as an imbalance between the two groups and is highlighted in bold.

**Table 4 ijerph-19-05335-t004:** Factors associated with changes in the use of a chronic medication due to COVID-19 among chronic medication users.

	Changed a Chronic Medication (*n* = 89)	Did Not Change a Chronic Medication (*n* = 2134)	cOR (95% CI)	aOR (95% CI) *
	N (%)	N (%)		
Depression score (EDS)				
EDS < 13	60 (67.4)	1175 (55.1)	Ref	Ref
EDS ≥13 (possible depressive symptoms)	29 (32.6)	959 (44.9)	0.6 (0.4–0.9)	**0.5 (0.3–0.9) ****
**GAD-7**				
0–4 (no anxiety)	26 (29.2)	960 (45.0)	Ref	Ref
5–9 (mild)	45 (50.6)	765 (35.9)	2.2 (1.3–3.6)	**2.3 (1.4–3.8)**
10–14 (moderate)	12 (13.5)	251 (11.8)	1.8 (0.9–3.6)	2.0 (1.0–4.2)
15–21 (severe)	6 (6.7)	158 (7.4)	1.4 (0.6–3.5)	2.0 (0.8–5.1)
Perception of COVID-19 severity during pregnancy/breastfeeding compared to non-pregnant/non-breastfeeding women				
Not more severe/not very risky/not risky et al	12 (13.5)	406 (19.0)	Ref	N/A
More severe/risky, very risky, extremely risky	64 (71.9)	1514 (70.9)	1.4 (0.8–2.7)	N/A
Changes in schedule of persons providing care (midwife, gynecologist)/cancellation or reduction in appointment				
No	56 (62.9)	1507 (70.6)	Ref	Ref
Yes	33 (24.7)	627 (29.4)	1.4 (0.9–2.2)	1.4 (0.9–2.2)
Coronavirus infection status ^1^				
Negative test and absence of symptoms	78 (87.6)	1870 (87.6)	Ref	N/A
Positive test or symptomatic	11 (12.4)	264 (12.4)	1.0 (0.5–1.9)	N/A
Close family member infected				
No	74 (83.2)	1763 (82.6)	Ref	N/A
Yes	15 (16.9)	371 (17.4)	1.0 (0.6–1.7)	N/A
Vaccine status				
Unwilling to be vaccinated	28 (31.5)	683 (32.0)	Ref	N/A
Vaccinated/willing to be vaccinated	46 (51.7)	1062 (49.8)	1.1 (0.7–1.7)	N/A
Anti-pandemic measures are an infringement of personal freedom				
Neutral	14 (15.7)	275 (12.9)	Ref	Ref
Agree/strongly agree	11 (12.3)	473 (22.2)	0.5 (0.2–1.0)	0.9 (0.5–1.6)
Disagree/strongly disagree	64 (71.9)	1386 (64.9)	0.9 (0.5–1.6)	0.5 (0.2–1.1)
Trust in various sources regarding the COVID-19 pandemic				
In either the media or the Internet	18 (20.2)	293 (13.7)	1.6 (1.0–2.8)	1.4 (0.8–2.5)
In either the government or health authorities	76 (85.4)	1714 (80.3)	1.2 (1.0–1.4)	1.4 (0.6–2.5)
In at least one health care professional ^2^	79 (88.8)	1820 (85.3)	1.4 (0.7–2.7)	N/A
In either family or friends	20 (22.5)	361 (16.9)	1.5 (0.9–2.4)	1.6 (1.0–2.8)

N/A = not applicable. cOR = crude odds ratio; aOR = adjusted odds ratio; CI = confidence interval; * Adjusted for country of response, comorbidity, EDS score, GAD-7 score, changes in schedule of persons providing care (midwife, gynecologist)/cancellation or reduction in appointments, anti-pandemic measures, trust in the media/Internet, trust in the government/health authorities, trust in family/friends; ** Adjusted odds ratio where the 95% CI does not include 1 are indicated in bold. ^1^ Anytime during the pandemic. ^2^ Health care professionals include general practitioners, pharmacists, midwives, and obstetricians.

## Data Availability

The collected data are presented in the manuscript and in the Appendix A.

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
