# Peer review of "Self-Reported Medication Use among Pregnant and Postpartum Women during the Third Wave of the COVID-19 Pandemic: A European Multinational Cross-Sectional Study"

_ijerph, 2022, doi:10.3390/ijerph19095335_

Round 1

Reviewer 1 Report

This study presents self-reported medication use among pregnant and postpartum women and during the third wave of the pandemic and if the are changes in chronic medication use due to COVID-19 and associated risk factors.

I appreciate the large number of data as well as the specification of Strengths and data limitations.

Considering that correlation is a measure of the strength of the relationship between 2 variables. I have a suggestion, to process the data to see if there are positive or negative correlations between the different variables.

The study is to be appreciated and certainly if it includes other EU member states, it will see changes related to self-medication among pregnant and postpartum women. The results reflect the medical situation in Norway (where most of the data comes from).

I noticed a possible writing error in line 227 “Of all participants, 5.X%”

And I recommend referencing the way of writing some bibliographic references no. 10 and 42.

Author Response

Reviewer 1:

This study presents self-reported medication use among pregnant and postpartum women and during the third wave of the pandemic and if the are changes in chronic medication use due to COVID-19 and associated risk factors. I appreciate the large number of data as well as the specification of Strengths and data limitations.

Point 1: Considering that correlation is a measure of the strength of the relationship between 2 variables. I have a suggestion, to process the data to see if there are positive or negative correlations between the different variables.

Answer 1: This is a very good point. We did test for multicollinearity between our variables using the variance inflation factor (VIF) with a threshold of 10 as indicative of multicollinearity but did not find any importantly correlated variables based on this threshold.

Point 2: The study is to be appreciated and certainly if it includes other EU member states, it will see changes related to self-medication among pregnant and postpartum women. The results reflect the medical situation in Norway (where most of the data comes from).

Answer 2: Indeed, we agree with the reviewer and have added a sentence in the limitations: "Nevertheless, our results should be interpreted in light of certain limitations. First, we observed an over-representation of Norwegian women, who tend to use less medication [32], compared to other countries."(line 466-468)

Point 3: I noticed a possible writing error in line 227 “Of all participants, 5.X%”

Answer 3: We thank you for pointing out this mistake, the correct percentage was 5.1% and has been changed accordingly (line 230).

Point 4: And I recommend referencing the way of writing some bibliographic references no. 10 and 42.

Answer 4: Indeed, it appears that these sources were not appropriately referenced and have been changed accordingly.                                                                                                                        Reference number 10 is now reported as: " The RECOVERY Collaborative Group. Effect of Hydroxychloroquine in Hospitalized Patients with Covid-19. N Engl J Med. 2020;383(21):2030-2040. doi:10.1056/NEJMoa2022926".                                                                                                         

Reference number 42 is now reported as : " Fatima T, Michael C, E Gerbier. Mental health of pregnant and postpartum women during the third wave of the Covid-19 pandemic – a European cross-sectional study. Under review.".

Reviewer 2 Report

Line 233: the authors report that women declare having taken a "medical product" but it is difficult to know whether it was a product traditionally consumed during this period (iron, folic acid, multivitamin, omega 3) or other medicines.

Line 256: again "at least one medication" is mentioned but the authors do not clarify whether it belongs to the group "other medicines (iron, folic, etc.)" or "other products".

If the authors divided the medicines into 6 groups, they should be more specific when presenting the results.

Line 269: What was the reason for not including NSAIDs in table 2 if it was the 2nd most used?

Discussion: Did the authors not investigate whether the decline in antibiotic use was influenced by the increased difficulty in accessing health services to purchase prescriptions?

Author Response

Reviewer 2:

Point 1: Line 233: the authors report that women declare having taken a "medical product" but it is difficult to know whether it was a product traditionally consumed during this period (iron, folic acid, multivitamin, omega 3) or other medicines.

Point 3: If the authors divided the medicines into 6 groups, they should be more specific when presenting the results.

Answer to points 1 and 3: We understand that the wording used to distinguish between the two groups may have caused some confusion since both medications and other medicinal products are grouped in our results as medicinal products. Thus, we changed the wording of other medicinal products to supplements/vitamins in the method as follows:  " National study coordinators classified each medicinal product into six different groups including "medications", "iron", "folic acid", "multivitamins", "omega-3 fatty acids" and "other products". The latter included probiotics, herbal products, vitamin C, calcium, and all other health products which did not belong to any other group. Iron, folic acid, multivitamins, omega-3 fatty acids and other products are referred to as "supplements/vitamins" in the text." (lines 142-147). We also added some precisions in brackets in the text: "Participants’ characteristics according to medication use (excluding iron, folic acid, multivitamins, omega-3 fatty acids and other products) in the last three months are presented in Table 1.  The use of supplements/vitamins (i.e iron, folic acid, multivitamins, omega-3 fatty acids and other products) by country is described in Supplemental Table 1." (lines 240-244).

Point 2: Line 256: again "at least one medication" is mentioned but the authors do not clarify whether it belongs to the group "other medicines (iron, folic, etc.)" or "other products".

Answer 2: We understand that this may cause some confusion in the reader and have added a precision in brackets as follows: " Overall, 41.4% (2'158/5'210) of women reported having used at least one medication (excluding supplements/vitamins) in the three preceding months [...]" (lines 265-266).

Point 4: Line 269: What was the reason for not including NSAIDs in table 2 if it was the 2nd most used?

Answer 4: Table 2 presents the ten most reported medication classes and individual medications among pregnant and post-partum women. In order to have a more concise table, we decided to present medications in order of descending prevalence among pregnant women for both groups. Thus, the corresponding prevalence for post-partum women do not necessarily correspond to the ten most frequent medication classes/individual medications within this group. We have added a clarification to the text: " Table 2 provides an overview of the ten most reported ATC classes and individual medications within each class among pregnant and postpartum women, in order of descending prevalence for pregnant women (see Supplemental Tables 1 and 2 for an overview by country). Thus, the medication classes and individual medications among post-partum women do not necessarily correspond to the ten most frequent ones within this group. For instance, NSAIDs do not appear in this table despite being the second most used class among postpartum women (16.7%). " (lines 275-282).

Point 5: Discussion: Did the authors not investigate whether the decline in antibiotic use was influenced by the increased difficulty in accessing health services to purchase prescriptions?

Answer 5: Indeed, this is a very interesting point. We suggested that the decrease of antibiotic use may have been caused by the reduced incidence of other infectious diseases due to the sanitary measures used to prevent covid-19. We could also hypothesize that women who suffered from common cold and might have sought medical care before the pandemic and perhaps received antibiotics at that time, chose not to consult. We mentioned the use of antibiotics in our first descriptive analysis since they were among the ten most used medication class in our study. However, our second analysis, which this time explored reasons for changes, focused on self-medication and chronic medication, which typically do not include antibiotics. This point should perhaps be explored in future studies.

Reviewer 3 Report

The paper by Gerbier et al. (manuscript IJERPH-1691386) represents a multicentric multinational retrospective cohort study performed on pregnant or postpartum women in order to investigate changes in self-administered medication patterns induced by the third wave of SARS-CoV-2 pandemics during the period June-August 2021. The study was well conceived and executed, the results were rigorously analyzed and succeeded to evidence several changes in medication profiles compared to similar studies performed during or before previous waves of the COVID-19 pandemics. The results are valid and of interest for a wide readership, therefore I recommend publication in International Journal of Environmental Research and Public Health in the present form.

Author Response

Reviewer 3:

Point 1: The paper by Gerbier et al. (manuscript IJERPH-1691386) represents a multicentric multinational retrospective cohort study performed on pregnant or postpartum women in order to investigate changes in self-administered medication patterns induced by the third wave of SARS-CoV-2 pandemics during the period June-August 2021. The study was well conceived and executed, the results were rigorously analyzed and succeeded to evidence several changes in medication profiles compared to similar studies performed during or before previous waves of the COVID-19 pandemics. The results are valid and of interest for a wide readership, therefore I recommend publication in International Journal of Environmental Research and Public Health in the present form.

Answer 1: We thank the reviewer for this comment and his/her appreciation of our paper.